# When Counterpoint Meets Chinese Folk Melodies

**Nan Jiang**[†]    **Sheng Jin**[†]    **Zhiyao Duan**[‡]    **Changshui Zhang**[†]
[†]Institute for Artificial Intelligence, Tsinghua University (THUAI),
State Key Lab of Intelligent Technologies and Systems,
Beijing National Research Center for Information Science and Technology (BNRist),
Department of Automation, Tsinghua University, Beijing, China
[‡] Department of Electrical and Computer Engineering, University of Rochester
jiangn15@mails.tsinghua.edu.cn    js17@mails.tsinghua.edu.cn
zhiyao.duan@rochester.edu    zcs@mail.tsinghua.edu.cn

## Abstract

Counterpoint is an important concept in Western music theory. In the past century, there have been significant interests in incorporating counterpoint into Chinese folk music composition. In this paper, we propose a reinforcement learning-based system, named FolkDuet, towards the online countermelody generation for Chinese folk melodies. With no existing data of Chinese folk duets, FolkDuet employs two reward models based on out-of-domain data *i.e.*, Bach chorales, and monophonic Chinese folk melodies. An interaction reward model is trained on the duets formed from outer parts of Bach chorales to model counterpoint interaction, while a style reward model is trained on monophonic melodies of Chinese folk songs to model melodic patterns. With both rewards, the generator of FolkDuet is trained to generate countermelodies while maintaining the Chinese folk style. The entire generation process is performed in an online fashion, allowing real-time interactive human-machine duet improvisation. Experiments show that the proposed algorithm achieves better subjective and objective results than the baselines.

## 1   Introduction

Counterpoint [31], as an important and unique concept in Western music theory, is commonly identified in the works composed in the Baroque and Classical periods. It refers to the mediation of two or more musical voices into a meaningful and pleasing whole, where the role of these voices are somewhat equal. Chinese folk melodies, on the other hand, are typically presented in a monophonic form or with accompaniments that are less melodic [28], with few exceptions [45]; they feature unique melodic patterns and are mostly based on the pentatonic scale. In the past century, some renowned Chinese composers, *e.g.*, Xian Xinghai and He Luting, have explored incorporating counterpoints and fugues to Chinese music [38] and these attempts have brought several successful choral and orchestral works such as "Yellow River Cantata" and "Buffalo Boy's Flute". However, systematic theories and broader influences on the general public of integrating counterpoint with Chinese folk melodies are lacking. This motivates our work.

In this paper, we propose a system named *FolkDuet* to automatically generate countermelodies for Chinese folk melodies, following the counterpoint concept in Western music theory while maintaining the Chinese folk style. Instead of offline harmonization, we make the generation in an online (causal) fashion, where the countermelody is generated in real time as the input melody streams in, for a broader research scope and better engagement of users [2]. We believe that this is an innovative idea and it would make broader impacts on two aspects: 1) It would facilitate music cultural exchanges between the West and the East at a much larger scale through automatic music generation and style

fusion, and 2) it would further the idea of collaborative counterpoint improvisation between a human and a machine [2] to music traditions where such counterpoint interaction was less common.

Automatically harmonizing a melody or a bass line into multi-part music has been investigated for decades, but primarily on Western classical music, especially J.S. Bach chorales. Early methods were rule-based [17, 43, 35], where counterpoint rules were encoded to guide the generation process. Later, statistical machine learning based methods were proposed to learn counterpoint interaction from training data directly. For Chinese folk melodies, however, both systematic counterpoint rules and large repertoires of countermelodies are lacking, making these existing approaches infeasible.

The key idea of our proposed method is to employ reinforcement learning (RL) to learn and adapt counterpoint patterns from Western classical music while maintaining the melodic style of Chinese folk melodies during countermelody generation. Specifically, we design two data-driven reward models for this task. The first one is trained on the outer parts (the soprano and the bass line) of Bach chorales to model generic counterpoint interaction. It models the *mutual information* between the outer parts of Bach chorales. The other rewarder is trained by *maximum entropy* Inverse Reinforcement Learning (IRL) on monophonic Chinese folk songs to model the melodic style. The reinforcement learning scheme then fuses the generic counterpoint interaction patterns with the melodic style during countermelody generation.

The proposed method works in an online fashion, aiming to support human-machine collaborative music improvisation. Compared to offline methods that can iteratively revise the generated music content (*e.g.*, using Gibbs sampling) [7, 19, 24, 47], online harmonization is more difficult to generate globally coherent and pleasing content. In this regard, the proposed method follows RL-Duet [27], which has shown that reinforcement learning helps to improve the global coherence in online harmonization in the style of Bach chorales. The proposed work differs from RL-Duet on two aspects: 1) It generates the countermelody in the Chinese folk style instead of the Bach chorale style on which the counterpoint rewarder is trained, and 2) it uses an iteratively updated rewarder (*i.e.*, IRL) instead of a fixed rewarder to achieve the knowledge transfer and style fusion.

To validate the proposed FolkDuet system, we compare it with two baselines: RL-Duet and a maximum likelihood estimation (MLE) baseline trained on outer parts of Bach chorales. Objective experiments show that FolkDuet generates countermelodies that show closer statistics to the Chinese folk melodies in the dataset and better key consistency with the melody. Subjective listening tests also show that the proposed approach achieves higher ratings than the baseline method, in terms of both harmonicity between the two parts and the maintenance of the Chinese folk style.

Our main contributions are summarized as follows: 1) To our best knowledge, this is the first attempt to fuse Western counterpoint interaction with the Chinese folk melody style in countermelody generation. 2) The proposed RL approach learns to generate countermelodies in real time for Chinese folk melodies purely from out-of-domain data (*i.e.*, Bach chorales, and Chinese folk melodies) instead of Chinese folk duets that are scarce. 3) The proposed counterpoint rewarder learns and transfers counterpoint interaction from mutual information between the two outer parts of Bach chorales to the Chinese folk countermelody task.

## 2 Related Work

### 2.1 Automatic Counterpoint Composition

There has been significant progress in automatic counterpoint composition. For example, David Cope developed the famous EMI (Experiments in Musical Intelligence) program [9], which employs elaborately designed musical rules to imitate a composer's style and create new melodies. Farbood *et al.* [12] implemented the first HMM algorithm to investigate the first-species counterpoint to write counterpoints. Herremans *et al.* [20, 21] used the tabu search to optimize the objective function involving constraints for counterpoint generation. J. S. Bach's chorales have been the main corpus in computer music for tackling full-fledged counterpoint generation. Several approaches have been proposed. Earlier work uses rule-based [3], constraint-based [39] or statistical models [8], while recent work mainly uses neural networks such as BachBot [34], DeepBach [19], Coconet [24, 25], and RL-Duet [27].

## 2.2 Reinforcement Learning for Music Generation

Recent works have explored reinforcement learning (RL) for automatic music generation [15, 6, 30, 18, 26]. RL-based approaches consider the long-term consistency and are able to explicitly apply non-differentiable musical rules or preferences. In SequenceTutor [26], hand-crafted rules are explicitly modeled as the reward function to train the RNN. However, it is difficult, if not impossible, to achieve a balance among these different rules especially when they conflict with each other. More recently, RL-Duet [27] was proposed to ensemble multiple data-driven reward models to get a comprehensive reward agent. Nevertheless, it still uses one rule to punish repetitions of generated notes.

In our work, we do not rely on hand-crafted rules or ensemble methods to obtain a comprehensive reward function. Instead, we use inverse reinforcement learning (IRL), which learns to infer a reward function underlying the observed expert behavior [37]. IRL and RL are usually trained alternately. such that the reward function could be dynamically adjusted to suit the reinforcement learning process. Such alternate training is similar to the generative adversarial networks (GAN). As some works draw an analogy between the theories of GAN and IRL [13, 22] from the optimization perspective, IRL could address the reward sparsity and mode collapse problems in GAN [42].

## 2.3 Unsupervised Style Transfer

Inspired by the recent progress of the visual style transfer [16], music style transfer is attracting more attention. Music style transfer is challenging because music is semantically abstract and multi-dimensional in nature [4]. According to [11], music style transfer can be categorized into 1) timbre style transfer for sound textures [44, 14], 2) performance style transfer for music performance control [36], and 3) composition style transfer for music generation [40, 32, 50, 29]. Regarding approaches, some methods achieve style transfer by disentangling the pitch content and rhythm styles in the latent music representations [40, 49], while others apply explicit rules to modify monophonic [50] or polyphonic melodies [29]. Our work can be viewed to transfer the Chinese folk style from the melody to the generated countermelody.

# 3 Methodology

## 3.1 Music Encoding

In our online countermelody generation task, each music piece has two parts: 1) the Chinese folk melody $H = [h_0, h_1, ..., h_{K_H}]$, which is composed by a human, and 2) the machine-generated countermelody $M = [m_0, m_1, ..., m_{K_M}]$ which harmonizes $H$. Here we use a note-based representation, and the $m_k$ and $h_k$ are the $k$-th note in the machine and human parts, respectively. As shown in Fig. 2(a), a note $m_k$ is represented with two items, $m_k^{(p)}$ and $m_k^{(d)}$ for pitch and duration, respectively. This is different from most existing work, which uses a grid-based representation with time being quantized into small beat subdivisions (*e.g.*, 16th notes [19]), and a note is then represented by a pitch-onset token followed by a sequence of hold tokens. In this work, we prefer a note-based representation for two reasons: 1) Note-based representations are closer to how humans compose and perceive music. 2) From the optimization perspective, note-based representations are more compact and do not have the token imbalance issue that grid-based representations have due to the much larger amount of hold tokens than pitch-onset tokens.

One challenge we face in our note-based representation is that it does not provide a natural synchronization between the two parts as grid-based representations do. To address this, we also encode the within-measure position $b_k$ of each note $m_k$. This $b_k$ is computed by modulating the note onset time $t_k$ by 16, assuming a 4/4 time signature and a position resolution of 16-th notes. The onset time $t_k$ uses metadata to make the generator have the concept of the beat. It is noted that although many melodies do not follow the 4/4 time signature, this simple approach achieves our goal of synchronization between the two parts.

## 3.2 Framework

For our task of online countermelody generation for Chinese folk melodies, we intend to achieve a globally coherent and pleasing interaction between the two parts borrowing the Western counterpoint

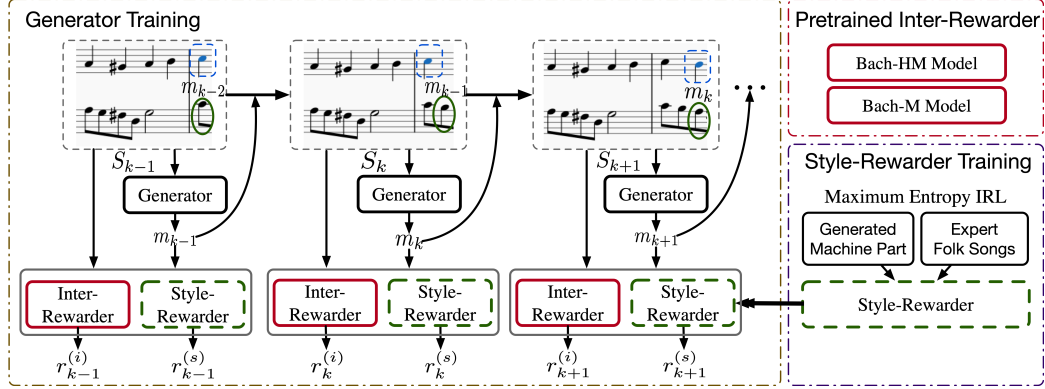

Figure 1: FolkDuet system framework. In each step $k$, the generator generates a new note $m_k$ in the machine part based on its observation of the state $S_k$, which contains all past notes input from the human and generated by the machine. Then the two rewarders evaluate $m_k$ based on their observations of $S_k$. The style-rewarder is trained alternately with the generator, using inverse reinforcement learning, while the inter-rewarder is pre-trained on Bach chorales and then fixed during the training of the generator.

concept, while maintaining the Chinese folk style in the generated countermelody. We propose to achieve this using reinforcement learning. Fig. 1 illustrates the framework of our proposed system, named *FolkDuet*. It contains a *generator* and two rewarders. The *inter-rewarder* models the counterpoint interaction in Western music, while the *style-rewarder* models the melodic pattern of Chinese folk songs. The training of the generator and the style-rewarder are alternated. The generator is trained through reinforcement learning with rewards provided by the two rewarders, while the style-rewarder is updated using the monophonic folk melodies in the training set (*i.e.*, the human part) and their generated countermelodies (*i.e.*, the machine part) using maximum entropy inverse reinforcement learning. Its learning objective is to infer the reward function that underlies the demonstrated expert behavior (*i.e.*, the human part). The inter-rewarder measures the degree of interaction between human and machine parts through a mutual information informed measure. This measure is computed by *Bach-HM* and *Bach-M* models, both of which are maximum likelihood models pre-trained on duets extracted from outer parts of Bach chorales. Different from the style-rewarder, the inter-rewarder is fixed during the reinforcement learning of the generator.

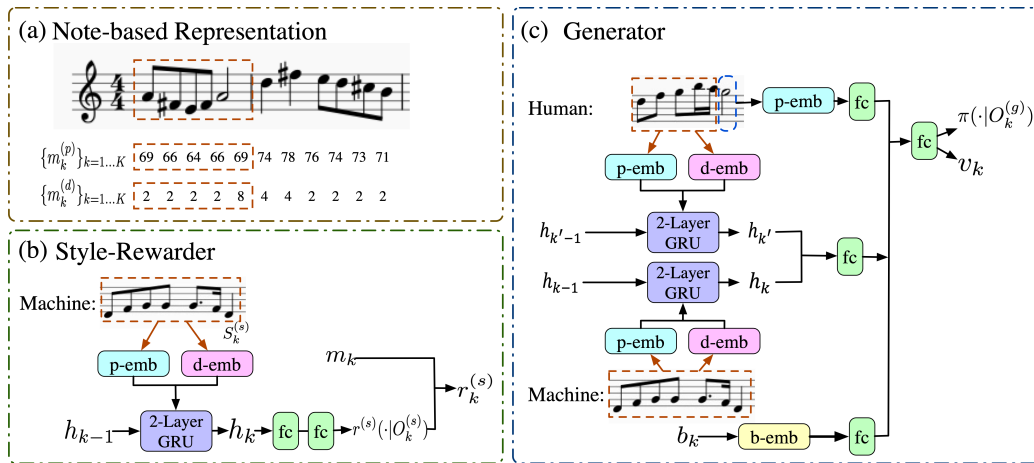

Figure 2: The note-based representation and the architectures of Generator and Style-Rewarder. The architectures of the two models in Inter-Rewarder are contained in the supplementary material. p-emb, d-emb and b-emb represent pitch/duration/beat embedding modules, respectively. GRU represents the Gate Recurrent Unit [5], and fc stands for the fully-connected layer.

## 3.3 Generator

The generator generates the next note $m_k$ using its policy $\pi(m_k|O_k^{(g)})$, where $O_k^{(g)}$ is its observation of the state $S_k$. Its training objective is to maximize the expected long-term discounted reward $\mathbb{E}_\tau R$. The long-term discounted reward of note $m_k$ is $R_k = \sum_{j=k}^{T+k} \gamma^{j-k}(r_j^{(s)} + \alpha r_j^{(i)})$, where $r_j^{(s)}$ and $r_j^{(i)}$ are the two immediate rewards for the future note $m_j$ from the style-rewarder and the inter-rewarder, respectively, and $\alpha$ is a weight factor.

In each training iteration, we first randomly pick one monophonic folk melody from the training data as the human part $H = [h_0, h_1, ..., h_{K_H}]$. Before generating the $k$-th note of the machine part, the generator's observation $O_k^{(g)}$ contains the already generated machine part $\{m_i\}_{i=0}^{k-1}$, the already completed notes in the human part $\{h_i\}_{i=0}^{k'-1}$, and just the pitch of the currently being played note in the human part $h_{k'}$. It is noted that the duration of $h_{k'}$ is not part of the observation, because in real-time human-machine interactive generation, the generator does not know when the note will end. The generator's observation strictly obeys the causality constraint. For notes in the human part that have ended strictly before the onset of the to-be-generated machine's note, it observes both the pitch and duration, while for the currently being played human's note, it only observes the pitch but not the duration. The human's note to be started at the same time as the machine note will not be observed whatsoever.

The architecture of the generator is illustrated in Fig. 2(c) and is detailed in the supplementary material. It outputs the probability of action $m_k$ conditioned on the current observation $O_k^{(g)}$, i.e., $\pi(m_k|O_k^{(g)})$, and the state value $v_k$. We use the actor-critic algorithm with generalized advantage estimator (GAE) [41] to train the generator.

## 3.4 Style Rewarder

The style rewarder aims to capture the Chinese folk melodic style. We employ the maximum entropy inverse reinforcement learning (IRL) [46] to infer a reward function underlying the demonstrated expert behavior, i.e., the Chinese folk melodies. As there are no existing data of Chinese folk duets, the Style-Rewarder is pre-trained on the monophonic Chinese folk melodies. We then use IRL and RL to alternatively update the Style-Rewarder and the generator. Following [46], the probability of a note sequence $\tau = \{n_k\}_{k=0}^K$ is proportional to the exponential of the reward along the trajectory $\tau$.

$$p(\tau) = \frac{1}{Z} \exp R(\tau), \tag{1}$$

where $Z = \int_\tau \exp\left(R(\tau)\right) d\tau$ is the partition function, $R(\tau) = \sum_{S_k, n_k \in \tau} r^{(s)}(O_k^{(s)}, n_k)$ is the accumulated rewards of path $\tau$ parameterized by the Style-Rewarder $R$, and $O_k^{(s)}$ is the Style-Rewarder's observation of state $S_k$. As in [42], to maximize the likelihood of the expert sequences (monophonic folk melodies), we have the following optimization gradient for the Style-Rewarder:

$$
\begin{aligned}
\nabla \mathcal{J}_r &= \mathbb{E}_{\tau \sim p_{\text{expert}}} \nabla \log p(\tau) \\
&= \mathbb{E}_{\tau \sim p_{\text{expert}}} \nabla R(\tau) - \frac{1}{Z} \int_\tau \exp\left(R(\tau)\right) \nabla R(\tau) \mathrm{d}\tau \\
&= \mathbb{E}_{\tau \sim p_{\text{expert}}} \nabla R(\tau) - \mathbb{E}_{\tau \sim p(\tau)} \nabla R(\tau) \\
&\approx \frac{1}{N} \sum_{i=1}^N \nabla R\left(\tau_i\right) - \frac{1}{\sum_j w_j} \sum_{j=1}^M w_j \nabla R\left(\tau_j'\right).
\end{aligned} \tag{2}
$$

The last step approximation replaces sampling $\tau \sim p(\tau)$ with sampling $\tau'$ from the generator, using importance sampling weight $w_j \propto \frac{\exp\left(R(\tau_j')\right)}{q_\theta(\tau_j')}$, where $q_\theta$ is the generation probability.

The architecture of the Style-Rewarder is illustrated in Fig. 2(b) and detailed in the supplementary material. Its observation $O_k^{(s)}$ of the state $S_k^{(s)}$ is only the already generated machine part. And it outputs the style reward $r_k^{(s)}$.

## 3.5 Inter-Rewarder

As we do not have access to Chinese folk counterpoint works, we use the paired outer parts of Bach chorales to learn a reward function about counterpoint interaction. We propose to measure this interaction by using mutual information, which has been introduced as an objective function that measures the mutual dependence [1, 33]. For two random variables $X$ and $Y$, their mutual information is defined as the difference of entropy of $Y$ before and after the value of $X$ is known, $i.e.$, $I(X,Y) = H(Y) - H(Y|X)$. It describes the amount of information shared between the two variables. A coherent counterpoint piece is expected to share an appropriate amount of information among its parts. The amount is neither too high ($e.g.$, identical parts) or too low ($e.g.$, irrelevant parts). The key to this rewarder is to know what amount is appropriate.

Let random variables $(X, Y)$ be a duet that follows the joint distribution of duets extracted from outer parts of Bach chorales, and let $X_i = \{x_k^{(i)}\}_{k=0}^{K^{x_i}}$ and $Y_i = \{y_k^{(i)}\}_{k=0}^{K^{y_i}}$ be the $i$-th sample from this Bach chorale duet dataset, then we have

$$
\begin{aligned}
I(X,Y) &= \sum_{X,Y} P(X,Y) \log \frac{P(X,Y)}{P(X)P(Y)} \\
&= \sum_{X,Y} P(X,Y) \left[\log P(Y|X) - \log P(Y)\right] \approx \sum_{X_i,Y_i \sim P_{X,Y}} \left[\log P(Y_i|X_i) - \log P(Y_i)\right] \\
&= \sum_{X_i,Y_i \sim P_{X,Y}} \left[\log \prod_{k=1}^{K^{y_i}} P(y_k^{(i)}|X_i, y_{t<t_k}^{(i)}) \cdot P(y_0^{(i)}|X_i) - \log \prod_{k=1}^{K^{y_i}} P(y_k^{(i)}|y_{t<t_k}^{(i)}) \cdot P(y_0^{(i)})\right] \\
&= \sum_{X_i,Y_i \sim P_{X,Y}} \sum_{k=1}^{K^{y_i}} \left[\log P(y_k^{(i)}|X_i, y_{t<t_k}^{(i)}) - \log P(y_k^{(i)}|y_{t<t_k}^{(i)})\right] + C(X_i, y_0^{(i)}).
\end{aligned}
\tag{3}
$$

The last item $C(X_i, y_0^{(i)})$ is constant for all the $\{y_k^{(i)}\}_{k=1}^{K^{(y_i)}}$. It can be seen that $\log P(y_k^{(i)}|X_i, y_{t<t_k}^{(i)}) - \log P(y_k^{(i)}|y_{t<t_k}^{(i)})$ is the immediate contribution to mutual information at the $k$-th step as the duet progresses in time. We trained two maximum likelihood models, Bach-HM and Bach-M, on the Bach chorale duets to learn $P(y_k^{(i)}|X_i, y_{t<t_k}^{(i)})$ and $P(y_k^{(i)}|y_{t<t_k}^{(i)})$, respectively. During the reinforcement learning of the Generator, we then define the interaction reward for the action $m_k$ using these models, as $\log P(m_k|H, m_{t<t_k}) - \log P(m_k|m_{t<t_k})$, where $H$ is the full human part, $m_{t<t_k}$ is the already generated machine part before the note $m_k$.

The intuition of this reward is to encourage actions $m_k$ that would result in a larger immediate contribution to mutual information as if the generated Chinese folk duet were to follow the Bach chorale distribution. Since the generated duets are far from this distribution, the rewarder is not trying to maximize the actual mutual information between the human and machine parts, which as argued earlier, should not be maximized anyway. Nevertheless, this reward somehow captures the counterpoint interaction.

## 4 Experiments

### 4.1 Experimental Settings

Two datasets are used in our work. The first one consists of Chinese folk melodies from the Essen Folksong Collection[1]. They are performance-feature-removed transcriptions, comprising 2250 traditional Chinese folk songs, from Han, Natmin, Shanxi, and Xinhua areas. Although many important performance-related features (e.g., glissandi) are not recorded in these transcriptions, we believe that they still convey the Chinese folk style to a good extent for our work. We filter out pieces that cannot be successfully read by music21 [10] or contain triplet notes. This remains 2022 folk songs. We randomly split them into 80% train, 10% validation, and 10% test. The other dataset is the Bach Chorale Dataset in music21. We only use the outer parts of the chorales to form duets, to train the Bach-HM and Bach-M models in Inter-Rewarder.

Each folk melody is transposed to all semitones such that its pitch range does not exceed that of the original dataset, which is from MIDI number 48 to 96. To address the pitch range mismatch between Bach chorales and Chinese folk melodies, we extend the pitch range of Bach chorales from [36-81] to [36, 96] by transposing each piece to all semitones within this range. There are 10 allowable durations of notes, calculated from the Bach chorales dataset, including 1/4, 2/4, 3/4, 1, 1.5, 2, 3, 4, 6, and 8 quarter-lengths.

The training process contains three stages: First, we train the Bach-HM and Bach-M models on the transposed Bach chorale duets. Second, we train initialization models for the Generator and the Style-Rewarder. To initialize the Generator, we create 20 pseudo folk duets for each folk melody in the training set, where the pseudo countermelody of each duet is simply a transposition of the main melody with a random pitch shift within the pitch range of the dataset. To initialize the Style-Rewarder, it is trained by maximizing the likelihood of the monophonic folk melodies. Finally, the generator and Style-Rewarder are trained alternately with reinforcement learning and inverse reinforcement learning, while the two Bach-HM and Bach-M models are fixed.

We compare FolkDuet with two baselines. The first baseline is a maximum likelihood estimation (MLE) model trained on our transposed outer parts of Bach chorales. This model needs to be trained on duets hence cannot be trained on the monophonic Chinese folk melodies in our training set. The second baseline is a pre-trained model RL-Duet [27], which is a recently proposed reinforcement learning algorithm for online duet counterpoint harmonization. It was again trained on Bach chorales, but the pitch ranges do not match those of the Chinese folk melodies. We did not compare with methods in [6, 15, 18, 26, 30], because models in [18, 26, 30] are designed for monophonic melody generation instead of harmonization, and rewards in [6, 15, 30] are rule-based and cannot appropriately capture the Chinese folk style or Bach chorale counterpoints. To address this pitch range mismatch issue, we first transpose each Chinese folk melody to a key within the pitch range accepted by RL-Duet before using RL-Duet to generate the countermelody, and finally transpose the generated duet back to the original key. For all of the three methods, we pick each folk melody from the test set, using its first 10 notes and the one octave transposed notes as the initialization of the duet. More information about the training process and hyper-parameters of FolkDuet and the baselines is contained in the supplementary material.

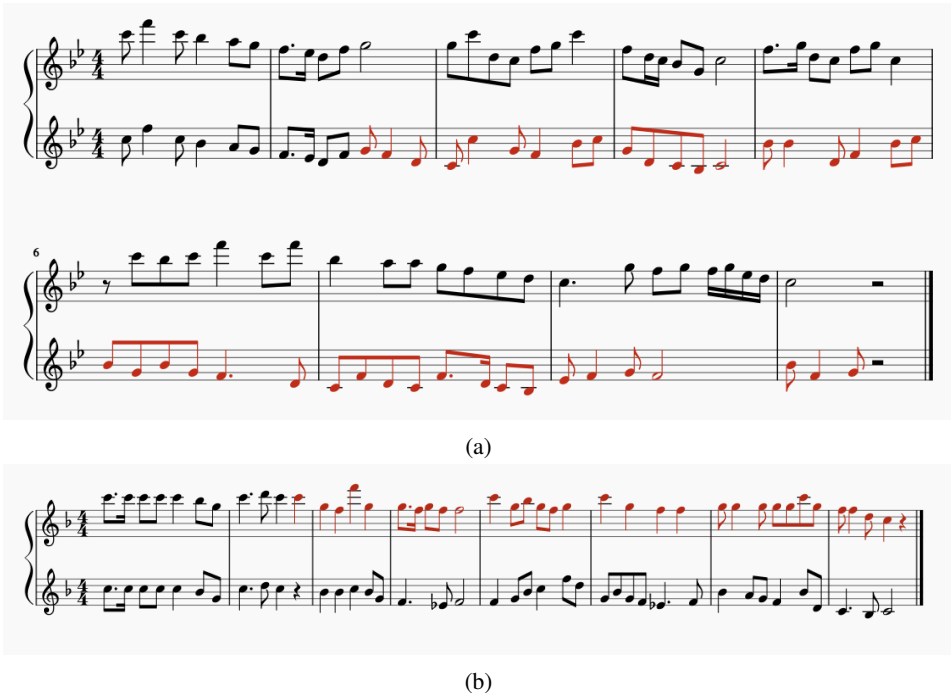

(a)

(b)

Figure 3: The generated countermelody is highlighted in red. We use the first 10 notes in the human part as its initialization, and these notes are transposed by either +12 or -12 semitones (randomly choosing one that does not exceed the pitch range of the dataset).

## 4.2 Examples

Fig. 3 shows two examples generated by FolkDuet. More examples are included in the supplementary materials.

## 4.3 Objective Evaluation

We adopt several metrics in [48] to evaluate whether the generated machine parts follow the Chinese folk style. Average pitch count per bar (PC/bar), pitch interval (PI), and inter-onset-interval (IOI) are scalar metrics, while pitch class histogram (PCH) and note length histogram (NLH) are vector metrics. For scalar metrics, we show the values in Table 1, while for vector metrics, we calculate the earth mover distance between the generated countermelodies and the original melodies in the test set. We can see that on all the metrics, countermelodies generated by FolkDuet achieve better style imitation than both baselines. Interestingly, the RL-Duet baseline shows statistics further from the dataset statistics than the MLE baseline does. This suggests that although reinforcement learning is effective in modeling counterpoint interaction [27], this modeling is difficult to adapt to a new music style if no style information is incorporated.

Table 1: Objective comparison results of FolkDuet and the two baselines (MLE and RL-Duet). For PC/bar, PI, and IOI, the closer to the dataset values, the better. For the other metrics, the arrow shows the better direction. The mean values and the standard deviations are calculated on 5 runs.

|  | PC/bar | PI | IOI | PCH ↓ | NLH ↓ | key-consist ↑ | inter-reward ↑ |
|---|---|---|---|---|---|---|---|
| Dataset | 3.90 | 2.73 | 2.36 | - | - | - | - |
| MLE | $4.21 \pm 0.12$ | $\mathbf{3.02 \pm 0.12}$ | $2.87 \pm 0.10$ | $0.017 \pm 0.002$ | $0.036 \pm 0.008$ | $0.78 \pm 0.01$ | $-0.30 \pm 0.02$ |
| RL-Duet [27] | $3.23 \pm 0.01$ | $4.02 \pm 0.01$ | $3.64 \pm 0.02$ | $0.017 \pm 0.001$ | $0.055 \pm 0.002$ | $0.71 \pm 0.01$ | $-0.50 \pm 0.004$ |
| FolkDuet | $\mathbf{3.96 \pm 0.12}$ | $\mathbf{2.44 \pm 0.14}$ | $\mathbf{2.16 \pm 0.10}$ | $\mathbf{0.008 \pm 0.001}$ | $\mathbf{0.014 \pm 0.004}$ | $\mathbf{0.85 \pm 0.01}$ | $\mathbf{0.13 \pm 0.03}$ |

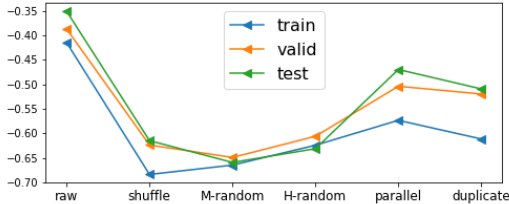

Figure 4: Interaction reward for different duets from the outer parts of the Bach's chorales.

The interaction between the two parts of a duet is harder to measure objectively. Nevertheless, here we adopt two simple metrics. One is the average key-consistency, which is defined as the cosine similarity of the pitch class histograms of the two parts. The other metric is the interaction reward used to train the generator. The last two columns of Table 1 show that FolkDuet outperforms both baselines significantly. To validate that this interaction reward can reflect counterpoint interaction, we calculate this reward on several kinds of the duets constructed from the outer parts of Bach chorales, including original duets (*raw*), duets of two randomly shuffled parts (*shuffle*), duets with random notes in the machine part (*M-random*), duets with random notes in the human part (H-random), duets of parallel human and machine parts (*parallel*), and duets of duplicate parts (*duplicate*). Fig. 4 shows that this interaction reward achieves the highest score on the original duets. The reason why FolkDuet has a higher interaction reward is due to $C(X_i, y_0^{(i)})$ in Equation 3, which is different for the distribution of the notes in the initialization segments of FolkDuet and Bach chorales.

## 4.4 Subjective Evaluation

We designed an online survey in Chinese to conduct listening tests. The survey starts with two music background questions.

1. *Have you learned to play a musical instrument / had vocal training / learned music theory, at least 2 hours a week for more than half a year in the past? -Yes -No*

2. *How much time do you spend in listening to music every day?*
   *- Less than 10 min. - 10 to 60 min. - More than 60 min.*

The survey then continues with paired comparisons between generated countermelodies from the MLE baseline and FolkDuet for the same melody, in a blind and random fashion. We exclude the RL-Duet baseline, as its Bach style is so strong that its generated parts differ sharply from the folk melodies. Some examples are shown in the supplementary. For the sake of resources, we excluded it in the listening test. A total of 184 pairs are randomly shuffled for each survey, and the participants are instructed to complete as many pairs as they wish. For each countermelody generation, the five-line music notation is rendered by MuseScore using the key label from the original melody and a 4/4 time signature. The subjects were informed to ignore the potential mistakes in pitch-spelling and to focus on listening. Audio recordings of the countermelody itself and the duet are rendered using the piano timbre. The music notation (with generated notes colored) and both audio recordings are presented to the participants. For all music notation in the listening test and examples in the supplementary material, we put the machine part at the bottom regardless of their pitch height (marked as red). After reading and listening to each generation, the participants are asked to rate the harmonic appealingness of the duet, the melodic appealingness, and the prominence of the Chinese folk style of the generated countermelody, all on a 5-point scale (from 0 to 4) from very weak to very strong.

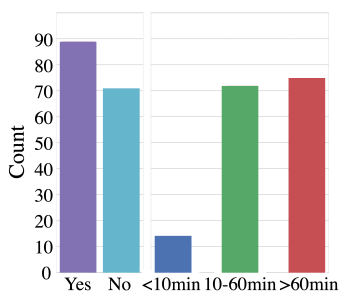 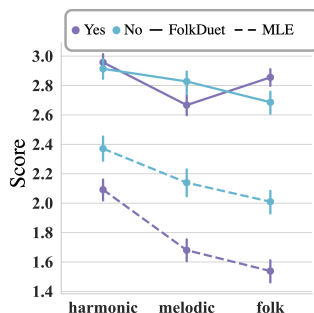 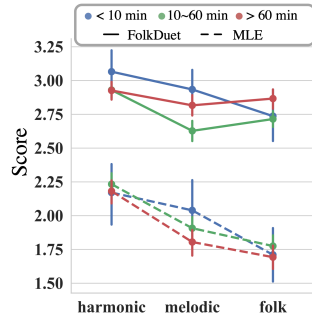

Figure 5: Background surveys of the participants.

Figure 6: Subjective scores organized by the background 1.

Figure 7: Subjective scores organized by the background 2.

In total, 160 people participated in the survey and 1,177 music pairs were compared and scored. On average each pair was scored by 6.4 subjects. Fig. 5 shows the music background distribution of the participants. Most participants have received a substantial amount of musical training. Figs. 6 and 7 illustrate the subjective scores. We observe that FolkDuet achieves higher scores in all three measurements than the baseline does. In particular, Fig. 6 shows that for participants with musical training ("Yes"), the score gap between FolkDuet and the baseline is larger, as the baseline scores all drop significantly from participants without musical training.

## 5 Further Discussion

In this paper, we made the first attempt to fuse Western counterpoint with Chinese folk style for countermelody generation for Chinese folk melodies. We proposed an online approach that is trained with reinforcement learning on Bach chorales and monophonic Chinese folk melodies. Subjective and objective evaluations show that our algorithm can generate harmonically and melodically pleasing and style consistent countermelodies for Chinese folk melodies.

We used two rewarders in Folk-duet. The Inter-Rewarder was held fixed while the Style-Rewarder trained with IRL. As the Chinese folk melodies for training are all monophonic, we could not adapt the Inter-Rewarder during RL training. Otherwise, the generator would lose the counterpoint ground in the style-counterpoint balance. When polyphonic Chinese folk training data is available, the Inter-Rewarder could also be trained through few-shot learning.

The scope of our application could be widened in both the music field and beyond music. First, counterpoint patterns could be transferred to other non-Western music styles. The generation could also contain more voices by evaluating the Inter-Rewarder between every two parts. Second, our work could be applied to compositions of other art forms, *e.g.*, classical Chinese poems, or couplets, which also have many counterpoint-like patterns.

# 6   Broader Impact

The idea of integrating Western counterpoint into Chinese folk music generation is innovative. It would make positive broader impacts on three aspects: 1) It would facilitate more opportunities and challenges of music cultural exchanges at a much larger scale through automatic generation. For example, the inter-cultural style fused music could be used in Children's enlightenment education to stimulate their interest in both cultures. 2) It would further the idea of collaborative counterpoint improvisation between two parts (*e.g.*, a human and a machine) to music traditions where such interaction was less common. 3) The computer-generated music may "reshape the musical idiom"[23], which may bring more opportunities and possibilities to produce creative music.

The proposed work may also have some potential negative societal impacts: 1) Similar to other computational creativity research, the generated music has the possibility of plagiarism by copying short snippets from the training corpus, even though copyright infringement is not a concern as neither folk melodies nor Bach's music has copyright. That being said, our online music generation approach conditions music generation on past human and machine generation, and is less likely to directly copy snippets than offline approaches do. 2) The proposed innovative music generation approach may cause disruptions to current music professions, even deprive them of their means of existence[23]. However, it also opens new areas and creates new needs in this *we-media era*. Overall, we believe that the positive impacts significantly outweigh the negative impacts.

# 7   Acknowledgments

This work is (jointly or partly) funded by National Key R&D Program of China (No. 2018AAA0100701), Beijing Academy of Artificial Intelligence (BAAI), and National Science Foundation grant No. 1846184.

## Footnotes

[1]https://kern.humdrum.org/cgi-bin/browse?l=/essen

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
