[Supplementary Material]

# Supplementary Material for
# When Counterpoint Meets Chinese Folk Melodies

## 1 Model Specification

### 1.1 Generator

Figure 1: The model architecture of Generator. p-emb, d-emb and b-emb represent pitch/duration/beat embedding modules, respectively. GRU represents the Gate Recurrent Unit [1], and fc stands for the fully-connected layer.

The generator's architecture is shown in Fig. 1. The history note sequences are highlighted with red dashed rectangles. They go through the pitch and the duration embedding modules and then are fed into the two-layer GRU one by one. Then the output of the GPU layer, the pitch embedding of $h_{k'}$ (marked with dashed blue rectangles), and the beat embedding of the onset time of $m_k$ all go through a fc layer and then concatenated. After one more fc layer, the generator gives a policy output $\pi(\cdot|O_k^{(g)})$ and a value $v_k$ output.

### 1.2 Style-Rewarder

The architecture of the Style-Rewarder is illustrated in Fig. 2. Its observation $O_k^{(s)}$ of the state $S_k^{(s)}$ is only the already generated machine part note sequences (marked with dashed red rectangles). The notes go through the pitch and the duration embedding modules and then are fed into the two-layer GRU one by one. The hidden state $h_{k'}$ is processed by two consecutive fully-connected layers. Finally, it outputs the style reward $r_k^{(s)}$.

### 1.3 Inter-Rewarder

The Inter-Rewarder consists of two models, *i.e.* BachHM and BachM.

Figure 2: The model architecture of Style-Rewarder. p-emb, d-emb and b-emb represent pitch/duration/beat embedding modules, respectively. GRU represents the Gate Recurrent Unit [1], and fc stands for the fully-connected layer.

The architecture of the BachHM model is demonstrated in Fig. 3. For BachHM model, the history and the future note sequences of the human part and the history note sequences of the machine part are processed by the pitch (p-emb) and the duration (d-emb) embedding modules, then the two-layer GRUs. The resulting hidden states are concatenated to get the context embedding. Again, we use fully-connected layers to map the current note embedding, the beat embedding, and the context embedding into $p(y_k^{(i)}|X_i, y_{t<t_k}^{(i)})$.

The architecture of the BachM model is shown in Fig. 4, which is the same as Style-Rewarder.

Figure 3: The model architecture of BachHM. p-emb, d-emb and b-emb represent pitch/duration/beat embedding modules, respectively. GRU represents the Gate Recurrent Unit [1], and fc stands for the fully-connected layer.

Table 1: Hyper-parameters of different Model architectures. The embedding size (of p-emb, d-emb and b-emb) and the hidden size (of GRU and fc layers) are listed.

|  | p-emb | d-emb | b-emb | GRU-nhid | GRU-nfc | p-nfc | b-nfc | pred-nfc |
|---|---|---|---|---|---|---|---|---|
| Generator | 128 | 64 | 32 | 256 | 512 | 128 | 32 | 512 |
| Style-Rewarder | 128 | 64 | 32 | 256 | 512 | - | - | 512 |
| BachM | 128 | 64 | 32 | 256 | 512 | - | - | 512 |
| BachHM | 128 | 64 | 32 | 128 | 256 | 128 | 32 | 512 |

Figure 4: The model architecture of BachM. p-emb, d-emb and b-emb represent pitch/duration/beat embedding modules, respectively. GRU represents the Gate Recurrent Unit [1], and fc stands for the fully-connected layer.

## 2 Training Process and hyper-parameters

The training process contains three stages: First, we train the Bach-HM and Bach-M models on the transposed Bach chorale duets. They are all trained for 20 epochs. Second, we train initialization models for the Generator and the Style-Rewarder. To initialize the Generator, we create 20 pseudo duets for each folk melody in the training set by transposing the melody with 20 random pitch shifts. To initialize the Style-Rewarder, it is trained by maximizing the likelihood of the monophonic folk melodies. These four models, are all trained for 20 epochs. We did a little hyper-parameter search of the GRU hidden size (128 or 256), the hidden size of the fc layer after the GRU (256 or 512), and the learning rate (0.01, 0.02, 0.05, 0.1). The best hyper-parameters are chosen according to the validation accuracy. Finally, the generator and the Style-Rewarder are trained alternately with reinforcement learning and inverse reinforcement learning, while the two Bach-HM and Bach-M models are fixed. We train the generator for 30 epochs with the entropy regularized actor-critic algorithm with generalized advantage estimator (GAE). And before training the generator for each epoch, the Style-Rewarder will be random initialized except the embedding layers and then trained for 4 epochs. They use the best model architectures, *i.e.* the GRU hidden size, the hidden size of the fc layer after the GRU, selected during the second training stage. We use dropout when training the initialization model for the Generator, however dropout rate is set to zero during the reinforcement learning, since the variance for the reinforcement learning is already enough large. We only tuned the learning rate (1e-5, 2e-5) and the Inter-Rewarder weight $\alpha$ (0.1, 0.5, 1) of the Generator. Finally, we choose the learning rate and the weight which receive generally best results on the objective metrics.

The final hyper-parameters are listed in Table 2, and the model architecture hyper-parameters are contained in Table 1.

Table 2: Hyper-parameters

| | | |
|---|---|---|
| Bach-HM | lr | 0.05 |
| | batch size | 256 |
| Bach-M | lr | 0.05 |
| | batch size | 256 |
| Style-Rewarder | initialization lr | 0.05 |
| | initilization batch size | 512 |
| | IRL lr | 1.00E-04 |
| | IRL batch size | 192 |
| Generator | initialization lr | 0.1 |
| | initilization batch size | 512 |
| | RL lr | 1.00E-05 |
| | RL batch size | 192 |
| | gamma | 1 |
| | entropy regularization weight | 0.05 |
| | IRL training sequence length | 20 notes |
| | Inter-Rewarder weight | 0.5 |

# 3 More results

## 3.1 Objective Results

Figure 5: Log-scaled pitch class histogram: Obvious differences exist between the test dataset and the generated samples of the baseline, *e.g.* the "D#" and "G#" pitch class. The mean values and the standard deviations are calculated on 5 runs.

Figure 6: Log-scaled note length histograms: Obvious differences exist between the test dataset and the generated samples of the baseline, *e.g.* 0.25 and 0.75 note length. The mean values and the standard deviations are calculated on 5 runs.

## 3.2 Subjective Results

Figure 7: Subjective Evaluation. The harmonic score, melodic score and folk style score. Higher is better. Our proposed FolkDuet outperforms the baseline significantly.

# 4 Examples

We compare several duet examples here. The generated countermelody is highlighted in red. We use the first 10 notes in the human part as its initialization, and these notes are transposed by either +12 or -12 semitones (randomly choosing one that does not exceed the pitch range of the dataset). The MP3 file are contained in the *examples* directory of the supplementary material.

# References

[1] Kyunghyun Cho, Bart Van Merriënboer, Caglar Gulcehre, Dzmitry Bahdanau, Fethi Bougares, Holger Schwenk, and Yoshua Bengio. Learning phrase representations using rnn encoder-decoder for statistical machine translation. In *The Conference on Empirical Methods in Natural Language Processing (EMNLP)*, 2014.

(a) FolkDuet

(b) Baseline

Figure 8: Example 1

(a) FolkDuet

(b) Baseline

Figure 9: Example 2

(a) FolkDuet

(b) Baseline

Figure 10: Example 3