[Reviews · NeurIPS 2020]

Review 1

Summary and Contributions: This paper proposes to use RL as a framework to create fusions in styles that are out-of-domain by transferring structure in one style of music onto another style where either that structure was less prominent or datasets that carry that structure were less available, while at the same time maintaining the characteristics of the other style. As a task, the paper proposes to generate counter-melodies to monophonic Chinese melodies by transferring counterpoint interaction from the outer voices of Bach chorales. The paper introduces FolkDuet, which consists of a generator and two reward functions. The generator takes in in an online fashion an existing folk melody and the counter-melody generated thus far and then generates the next "counter note". The two reward functions include, a "inter-rewarder" that uses mutual information to capture the counterpoint relationships in Bach chorales by pre-training on the top and bottom voice in Bach chorales, and a "style-rewarder" that captures the melodic style, in this case pre-trained/initialized on monophonic Chinese melodies. The training interleaves the generator being trained with the actor-critic algorithm on the current reward functions with training the "style-rewarder" with inverse RL, while the "inter-rewarder" is held fixed. The paper shows improvements over baselines that are not able to consider two styles at once, both on musical metrics and human listening tests.

Strengths: The paper proposes a highly novel approach to combining styles in a creative way without needing ground truth data. It gives a technical framework for learning to trade-off and to balance preserving characteristics in one style while being able to augment it with structure from another style. The proposed components of the approach align well with the challenges of the task and domain, i.e. using mutual information to capture and transfer structure and IRL to take advantage of expert demonstration that bears characteristics for the parts to be added where there is not yet data for (i.e. the counter-melodies). The paper provides a compelling example of how the challenges in a domain motivated a new formalization to a potentially more general problem. The generated samples sound convincing. The melodies preserve the folk melody style while the counterpoint carries nice contrary motion, and the rhythmic movement of the two lines complement each other.

Weaknesses: The scope of the application could be widened. The paper only focuses on one application of transferring the outer-voices of Bach chorales to generating counter melodies to monophonic Chinese melodies. Are there other tasks within these styles of music or fusions of other styles of music that could be considered? For example, is it possible to include not just the other voices but more voices? As this would introduce more of the harmonic structure in the Bach chorales, does this require the inter-rewarder to be adapted during training too? Does it require additional sources of data that shows how Chinese melodies can be harmonized with common-practice chords. Additionally, since the musical domain is a core part of the paper, it could be helpful to provide a slightly deeper introduction to the musical traditions, especially in how "counterpoint" is approached in Chinese music. This could also inform and motivate why Bach chorales might be appropriate for augmenting the duet structure in Chinese music. Related to the broader impact section of the paper, the discussion could include the opportunities and challenges of cultural exchange in both directions. What are some potential applications and impact beyond music?

Correctness: As the paper tackles a new domain, there are less specific prior work to compare to. For example, prior work RL-Duet only learns from one style. Currently both of the baselines focus on learning from the Bach chorales, are there other baselines that could be constructed using simple heuristics that leverage the folk styles? This can also illustrate the difficulty of the tasks. One naive proposal of a heuristic could be to generate the counter melodies by chunking existing folk melodies and then matching them to existing melodies as a counter part by taking pitch/key statistics, contrary/parallel motion, rhythm etc into consideration.

Clarity: The paper makes several domain and technical contributions but the title only describes the domain setup. The paper is overall well-written. Some additional clarifying questions below. - There could be more discussion on why one of the rewarder is held fixed while the other is allowed to adapt. What are the implications on the trade-off between "melodic" versus "harmonic" structure? - Line 152, Does the observations for the generator that's about to generate the kth note include the kth note being played by the "human" part? Would that violate the causal online constraint? - Section 3.4 could be restructured by first explaining the context and setup more. For example, in line 167, to “maximize the likelihood of the expert sequence from the dataset”, does the expert sequence refer to the monophonic Chinese folk melodies from the dataset? In line 217, it is mentioned that the style-rewarder is first pre-trained with the monophonic Chinese folk melodies. This could be mentioned here in section 3.4 and also Figure 1 to clarify how the expert sequences come into play. In the evaluation section: - Line 217, what kinds of intervals does the 20 random pitch shifts include? - Line 246, what does “interaction reward” in the context of measuring “interaction between the two parts of a duet” refer to? Does it refer to the “inter-rewarder” or the both the “inter-rewarder” and “style-rewarder”? If only the “inter-rewarder” is included, there could be more discussion on why FolkDuet has a higher interaction reward than the original Bach chorales. - In reference to line 267, there could be some discussion on the objective of including music notation in the listening test. How does that affect the listeners and non-musicians perception? Since pitch is represented as MIDI numbers in the training data, how is pitch spelling handled for creating the scores? Would a “mistake” in pitch spelling influence participants' perception? - In Figure 7, it’s odd to see the higher melody on the bottom. Is the model supposed to generate a lower voice as a counter melody? Or is the generated output transposed?

Relation to Prior Work: Yes, it is clear how this work differs from RL-tuner and RL-Duet.

Reproducibility: Yes

Additional Feedback: Thank you authors for the thorough feedback. It clarifies my questions. This paper presents a compelling approach to style transfer/fusion on a structural level, leveraging the multifaceted kinds of structure in music. I believe this paper will inspire many future work and have high impact for creative ML hence I'm increasing my overall score from 6 to 7.


Review 2

Summary and Contributions: This work looks at the application of generative modeling to musical style transfer. In particular, they look at the novel application of fusing the musical styles of Wester Counterpoint and Chinese folk melodies. The proposed approach to this problem relies on neural network generative model trained on two task-specific reward functions. The first reward function rewards pieces where the duet have mutual information (grounded in pretrained models), while the later is an inversely-learned reward function that tries to match the style to expert demonstrations. The performance is measured both quantitatively across many metrics, and by a survey.

Strengths: This work provides a clear introduction into the application of generative models for musical composition. They are careful to introduce slowly musical jargon, and explain terms such that machine learning experts may find them accessible. The clearness of definition continues on through they proposed methodology where they carefully explain each component to their system, and motivate why this is important from both a musical and learning perspective. This work also tackles the under-studied problem of musical generative models. I found the approach to their design of reward functions to be interesting and a clever application.

Weaknesses: The area for the largest potential improvement is in the subjective evaluation. Their survey instrument contained two questions to contextual the expertise of the surveyee, and then asked for evaluations of “harmonic appealingness, melodic appealingness, and prominence of the Chinese folk style”. While the evaluation questions are reasonable on their own (I would encourage the authors to also collect qualitative feedback in the form of open ended questions); I think the background questions do not get a solid picture of the surveyee’s knowledge. The first question critically asks if the surveyee has had training in music; however, does not check the recency of the training. Therefore, the knowledge may no longer exist. The second question asks about music listening habits; however, I would suspect this a weak confounder to expertise. To exemplify, I listen to a lot of music; however, would not self-identify as having any music theory knowledge. All this being said, the survey results are still overwhelmingly in favor of the proposed work. Additionally, I would prefer it if the authors spent more time motivating why this particular application is worthy of attention to a machine-learning audience. While I can appreciate the ingenuity of the architecture’s design; it’s not clear from the paper alone why Counterpoint and Chinese-folk are an application worth studying. The author’s attempt this by justifying it in broader strokes: human-computer composition, and bringing cultures together. However, it would make the work stronger to either justify this particular application or show that this solution works across different style-fusions -- showing that it isn’t extremely limited in application.

Correctness: The correctness of the machine learning contributions are correct. I am not qualified to verify that the musical theory is sound.

Clarity: The paper is very well written; it should be very approachable for a machine learning audience with little musical background. Small nits: - Line 9: Awkward “that form good a counter part” - Table 1: typo: ad → and - Table 1: why is MLE’s PI bolded?

Relation to Prior Work: There is a clear discussion about pre-existing work and how it differs from the proposed FolkDuet.

Reproducibility: Yes

Additional Feedback: I appreciate the inclusion of the samples in the supplementary material.


Review 3

Summary and Contributions: The paper presents a method based on RL-Duet to generate counterpoint in real-time on transcriptions of Chinese folk music melodies. The contributions can be summarized as: - Use inverse reinforcement learning which learns the reward from the data - Novel application

Strengths: - Builds upon prior work with clear improvements shown on the dataset - Novel usage of two different music style (there are also some problems in this sense, see weaknesses) - Rigorous evaluation (except DuetRL is not included in the subjective evaluations, see weaknesses)

Weaknesses: - Even if MLE was better than DuetRL in objective evaluations, it is not a reason to leave DuetRL out of subjective evaluations. - The musical motivation and background are explained weakly. There are no musicological sources cited for counterpoint (lines 15-17) and Chinese folk music (lines 18 - 20). We do not know the composers who use incorporate counterpoint to Chinese folk melodies or their prominence (line 20). Are these compositions Western classical music compositions, part of another classical tradition (Beijing opera?), or something else (film music?)? As far as I understand, variations of Chinese folk music have been a common theme throughout the development of Western classical style in China since the 20th century (e.g., He Luting's compositions)? Is there a demand to create more "counterpointed" folk music, e.g., for music education, film/game music generation, for commercials? In short, The readers should be able to appreciate these details in the introduction and be able to read further from additional references. - The quality and reusability of Chinese folk music transcriptions are not discussed. I think that the dataset is fine however, the usage of this dataset has to be justified by the authors, at least part of the supplementary. See the next three comments below: 1) Folk music is transmitted orally; scores are transcriptions into a single melody. These transcriptions do not typically include heterophony and other stylistic features like glissandi, ornaments, dynamics contrasts, etc. that characterize the folk song genre or style. The authors should discuss the limitations of the lack of performance elements in the transcriptions. 2) Another important aspect is the collection process and purpose of the transcriptions and if they are suitable for music generation? Given that the data is retrieved from Essen folk collection, is it possible that European scholars who prepared transcriptions with a Western classical music perspective and without adequate understanding of the characteristics of Chinese folk music? Are we sure that the transcription process retained the melodic characteristics? Such transcriptions are sometimes useful for descriptive reasons from a "Western" point of view, and they are not intended for prescriptive work, e.g., to be used in performance practice. (https://www.amherst.edu/system/files/media/1770/Seeger%252520-%252520Prescriptive%252520and%252520Descriptive%252520Music-Writing.pdf). There is no discussion if these transcriptions are re-usable for rendering music. To the authors' defense, I explored the dataset and Essen archives for more evidence (https://wiki.ccarh.org/wiki/EsAC) and consulted an ethnomusicologist specializing in Chinese music to validate the dataset. The response was that the transcriptions sound OK albeit that lack the performance elements as described above. Nevertheless, - since it is going to be the first paper on the topic - the authors need to include these discussions as part of the study (paper or the supplementary). 3) There are many different Chinese folk music traditions, which significantly differ in characteristics due to region, ethnicity, and era. This is not acknowledged by the authors, except in Section 4.1: "It comprises 2250 traditional Chinese folk songs, from Han, Natmin, Shanxi, and Xinhua origins." However, the origins are confusing: Han is an ethnicity; Shanxi is a province; Xinhua is a small county or the official state-run press agency, and the transcriptions listed under Xinhua are from all over China and modern-day Mongolia; I am not sure what Natmin is. This is a sloppy explanation from the authors' side.

Correctness: Apart from the musicological parts, the claims are valid and the method is sound. DuetRL should be included in the subjective evaluation.

Clarity: Overall it is well-structured and written however, there is still a room for improvement: The meaning of "outer part of a chorale" (I assume it is either soprano or bass line) is not explained explicitly, which would make it difficult for any reader without advanced Western classical music theory training to understand. In Section 4.1 second paragraph, the narration jumps from melody range to duration and then back into range, which is confusing. Line 37: "making these existing approaches do not apply." => "making these existing approaches infeasible." Line 107: The sentence a bit difficult to read, it might be better to rewrite, e.g.: "1) the melody, H = [...], which is performed by a human, and 2) the machine-generated countermelody, M = [...], which harmonizes H." In addition, it might be better here (rather than line 135) to explicitly state that the "human melody" is a Chinese folk melody, for clarity's sake. - Line 176: As we do not have access to Chinese folk counterpoint works, => This sentence is misleading. As the authors also state in the paper (it's one of their motivations), counterpoint does not exist in the oral discourse of Chinese folk music. You cannot have access to something that does not exist. - Line 239: "achieve show" double verb - Line 246: "outperfors" -> outperforms - Section 4.4. would be better after Section 4.1. - References has several problems: - Conference names are mostly abbreviated, please write the full name. - Some words lack capitalization, e.g. "lstm" in [29], "arxiv" in [27] - [34] is a blog post (https://ashispati.github.io/style-transfer/), please cite it correctly

Relation to Prior Work: Yes. Section 1 and 2 covers prior work and the contributions of the paper clearly.

Reproducibility: Yes

Additional Feedback: - Beat synchronization paragraph (Line 119 - 125). While acknowledging that most pieces are not in 4/4, they assume that a 4/4 time signature, infer the beats from the metadata, and divide the intervals into 16. They write that "this simple approach achieves [the] goal of synchronization between the two parts." To me, the explanation is not clear. They do not write clearly which metadata they use or how they translate the metadata into beat locations. They can read the meter information kern score headers in the first place, so I am not sure why they assume 4/4. I think this part requires rephrasing. - Line 297: Neither folk tunes (anonymous) nor Bach's music (centuries-old) has any copyright. Therefore the first negative impact cannot happen. In this case, copyright infringement could only occur if FolkRL generates an already copyrighted melody by chance. - The impact part can be improved by referring to existing literature: see Example 1 in Holzapfel, A., Sturm, B. & Coeckelbergh, M. (2018). Ethical Dimensions of Music Information Retrieval Technology. Transactions of the International Society for Music Information Retrieval, 1(1), 44-55. (https://transactions.ismir.net/articles/10.5334/tismir.13/galley/14/download/). ========== Edit post-rebuttal: I would like to thank the authors for addressing the bulk of my concerns. I am happy with the authors' responses on improving 1) the musical background, 2) the description and shortcomings of the score representation and the dataset, and 3) the impact/ethics section. I hope the paper will be accepted for publication by the conference committee, and I look forward to reading the final version of the paper in the proceedings.


Review 4

Summary and Contributions: This paper uses reinforcement learning to harmonize Chinese folk melodies in a style influenced by the tradition of Western counterpoint. Quantitative metrics and subjective listening tests show that the proposed method produces more pleasing countermelodies than previous work, while better conforming to the distribution of the Chinese folk melodies in the dataset.

Strengths: The paper is well written, and the experimental design is clear and well executed. The examples in the supplementary material are interesting, and the contribution of this application is novel and well motivated.

Weaknesses: I wondered while reading through this whether simpler baseline approaches could be competitive with your RL approach. The MLE baseline and the previous RL work are good comparisons to make, but the context of this countermelody composition problem made me wonder whether there might be simpler ruled-based or grammar-learning approaches that could work well. Though I don’t expect you will have time to add more experiments to this paper, some discussion of why you felt that the methods mentioned in section 2.2 were not were comparing against in your study would be helpful.

Correctness: Yes

Clarity: Yes

Relation to Prior Work: Yes

Reproducibility: Yes

Additional Feedback: Interesting application, and I’m curious to see where this goes. I agree that the benefits have the potential to outweigh the downsides. -- Update Thanks for your response, I look forward to seeing the final paper.

[Author Response · NeurIPS 2020]

**We thank all reviewers for the constructive comments! We now respond to common and individual comments.**
**Common:** Q1: On musical motivation and background. In the revision, we will give more introduction and references.
A brief one: Counterpoint [1] is an essential and unique concept in Western music theory. Traditional Chinese music
(*e.g.*, folk songs[2] and operas), in its native form, does not have counterpoint practices, with few exceptions [3]. Some
renowned Chinese composers, *e.g.*, Xian Xinghai and He Luting, have explored incorporating counterpoints and fugues
to Chinese music [4]. Notable works include "Yellow River Cantata" by Xian Xinghai and "Buffalo Boy's Flute" by He
Luting. However, systematic theories and broader influences on the general public of integrating counterpoint with
Chinese folk melodies are lacking. This motivates our work. [1] Q2: On the scope of application. Yes, the scope of the
application could be widened in both (1) the music field and (2) beyond music in our future work. (1) Counterpoint
patterns could be transferred to other non-Western music styles. The generation could contain more voices by evaluating
the inter-rewarder between every two parts. (2) Our work could be applied to composition of other art forms, *e.g.*,
classical Chinese poems or couplets, which are rich of counterpoint-like patterns. Q3: On broader impact. In the
revision, we will discuss more opportunities and challenges of cultural exchanges in both directions. For example, the
inter-cultural style fused music could be used in Children's enlightenment education to stimulate their interest in both
cultures. We will also discuss other ways in which MIR may change music, following the suggested reference.

■ **Reviewer 1**: Q1: On the baseline. Thank you for the suggestion on building another baseline with heuristics and
rules. However, we find it not trivial to meet our online generation setting, and plan to attempt it in future work.
Q2: On whether the rewarder is fixed. As the Chinese folk melodies for training are all monophonic, they are not
appropriate to train or fine-tune the inter-rewarder; Adapting the inter-rewarder during RL training is likely to lose
the counterpoint ground in the style-counterpoint balance. When polyphonic Chinese folk training data is available,
we plan to adapt the inter-rewarder through few-shot learning. Q3: On causality. The generator's observation strictly
obeys the causality constraint. Line 152-153 and Figure 2 both show that: the generator will only observe the pitch and
duration of the notes that end strictly before the onset of to-be-generated note, and only the pitch (but not duration) of
the currently being played note. Q4: On the 20 random pitch shifts. The 20 pseudo duets for initializing the generator
training are obtained by first randomly transposing the folk melody 20 times, and then creating a pseudo counterpart
for each transposition through another random transposition. All transpositions cannot exceed the pitch range of the
dataset. Q5: On the interaction reward. Yes, it refers to the "inter-rewarder". The reason why FolkDuet has a higher
interaction reward is due to the $C(X_i, y_0^{(i)})$ in Equation (3), which is different for the distribution of the notes in the
initialization segments of FolkDuet and Bach chorales. Therefore, a high interaction reward does not necessarily
mean a higher mutual information. Q6: On music notation in listening test. The music notation was shown to help
subjects to attend the two voices. We used the notated key of the human part and the 4/4 time signature to render
pitch spelling and bar line positions. There were some mistakes and subjects were informed to ignore such potential
mistakes and focus on listening. Q7: On staff ordering in Figure 7. The model can generate both lower and higher
voices. For all music notation in the listening test and examples in the supplementary material, we put the machine
part at the bottom regardless of their pitch height. ■ **Reviewer 2**: Q1: On background questions in listening test.
Thank you for pointing this out and we admit that the background questions could be designed more carefully. In
future work, we will survey subjects' recency of music training. Q2: On writing. Thanks! We will improve these
in the revision. ■ **Reviewer 3**: Q1: On RL-Duet in listening test. We considered adding RL-Duet in listening test.
However, its Bach style is so strong that its generated parts differ sharply from the folk melodies. So, for the sake of
resources, we excluded it in listening test. In the revision, we will make this clear in paper and show some examples
in the supplementary and website. Q2: On quality and reusability of Essen dataset. Thank you for the suggestions!
We admit that there are many performance-related features in Chinese folk music that cannot be recorded in the MIDI
transcriptions. That being said, the typical transcription format of Chinese folk music is the numbered musical notation
(plus occasional playing technique marks such as glissandi), which is almost equivalent to MIDI. This suggests that
these performance-feature-removed transcriptions still convey the Chinese folk style. Therefore, we believe that our
symbolic generation work is well grounded on this dataset. In the revision, we will provide a clearer description and
justification. For the Essen dataset, the "Han, Natmin, Shanxi, and Xinhua" classification comes from the original
dataset and also confuses us. In the revision, we will add a footnote and refer readers to the original dataset for more
information :) Q3: On beat synchronization. We calculate beat locations of notes with the 4/4 time signature assumption
and 16th note quantization. Take notes in Figure 2(a) as an example, their onsets are, in 16th note indexes, at 0, 2, 4, 6,
8, 16, 20, 24, 26, 28 and 30. Their beat locations are then at 0, 2, 4, 6, 8, 0, 4, 8, 10, 12 and 14. As many pieces in the
Essen dataset use other time signatures and may even vary them within a piece, we admit that our simple treatment may
confound the model training, and we plan to improve this in future work. Q4: On writing. Thanks! We will fix them all
in the revision. ■ **Reviewer 4**: On the baseline. We did not compare with methods in [5, 14, 17, 24, 27], because (1)
models in [17, 24, 27] are designed for monophonic melody generation instead of harmonization, and (2) rewards in [5,
14, 27] are rule-based and cannot appropriately capture the Chinese folk style or Bach chorale counterpoints.

of Polyphonic Treatment of Chinese Tunes. [4] Ouyang, et al. Ideology and National Identity in 20th-Century Chinese Music.

## Footnotes

[1][1] Laitz. The complete musician. [2] Jones, et al. Folk music of China: Living instrumental traditions. [3] Wiant. Possibilities


[Meta-Review · NeurIPS 2020]

All reviewers recommend acceptance. They appreciated the novelty and ingenuity of the approach, and the quality of the resulting samples. The key limitation of this work is the relatively narrow scope, however, I do not believe that should be grounds for rejection, given the quality of the work. There were also some concerns about the dataset and the evaluation setup, but these have been adequately addressed in the author feedback. Please make sure to update the manuscript in line with the commitments made in the author feedback.